# Spearfishing and public health promotion: A cross-sectional analysis of the Hawai'i Behavioral Risk Factor Surveillance System Survey

**Lauryn Hansen**[1,2], **Yan Yan Wu**[1], **Tetine Lynn Sentell**[1], **Mika Thompson**[1], **Tonya Lowery St. John**[1], **Simone Schmid**[1], **Catherine McLean Pirkle**[1]*

1 Office of Public Health Studies, Thompson School of Social Work and Public Health, University of Hawai'i at Mānoa, Honolulu, Hawai'i, United States of America, 2 University of Hawai'i Sea Grant College Program, University of Hawai'i at Mānoa, Honolulu, Hawai'i, United States of America

* cmpirkle@hawaii.edu

## Abstract

Spearfishing, a culturally relevant practice in many locations globally, may foster physical activity and enhance well-being by promoting social cohesion, food security, and nature connectedness, but is understudied in public health promotion and surveillance. This study measured the population-level prevalence of lifetime spearfishing engagement in Hawai'i and identified associated factors for public health promotion. The Hawaiian Islands present an ideal setting for such activities due to its central Pacific location and a diverse population with cultural ties to spearfishing. In 2019 and 2020, lifetime spearfishing engagement was added to the Hawai'i Behavioral Risk Factor Surveillance System (N=12,737). Prevalence ratios (PR) and 95% confidence intervals (95%CI) were estimated for spearfishing "sometimes," "often," or "very often" during one's lifetime, considering sociodemographic, health behavior, and health status variables. A quarter of respondents statewide reported engagement, with higher rates amongst men (41%), Native Hawaiians (43%), other Pacific Islanders (36%), American Indian or Alaskan Native (32%), and rural island residents of Lāna'i (51%) and Moloka'i (43%). All age groups reported similar lifetime engagement. After statistical adjustment, those with a high school diploma or less were significantly more likely to have engaged in spearfishing than those with higher education. Spearfishing engagement was also associated with a higher likelihood of meeting physical activity guidelines (PR 1.45 95%CI 1.29-1.63). There is widespread lifelong engagement in spearfishing in Hawai'i, especially among Indigenous and rural populations. Supporting culturally relevant activities, such as spearfishing, is a strength-based approach to health promotion with global relevance, including encouraging physical activity.

## Introduction

Physical activity plays a crucial role in promoting overall health and well-being. However, research suggests that only 28% of adults worldwide and 24.8% of adults in Hawai'i meet the current physical activity guidelines [1,2]. Physical inactivity is defined as not meeting the recommended 150 minutes of moderate-intensity exercise or 75 minutes of vigorous-intensity exercise per week.

**Data availability statement:** The data have been made available in the Supporting information.

**Funding:** This work was supported by the Hawaiʻi Department of Health, Chronic Disease Prevention & Health Promotion Division, through a contract with the University of Hawaiʻi at Mānoa. Catherine Pirkle and Tetine Sentell were the recipients of the award. No commercial companies were involved in funding the work; no individual received funding from commercial companies for the purpose of this project. There was no additional external funding received for this . The Hawaiʻi Department of Health is responsible for the Hawaiʻi Behavioral Risk Factor Surveillance System. Catherine Pirkle and Tetine Sentell successfully negotiated with the Hawaiʻi Department of Health to include a question on spearfishing engagement during the 2019 and 2020 waves. Per national standards for the Behavioral Risk Factor Surveillance System, data collected from this system are public. Any data from the Hawaiʻi specific questions is available to the public following proper data request protocols submitted to the Hawaiʻi Department of Health. This includes the question on spearfishing engagement. The views expressed in the paper are the sole content of the authors and do not represent the official views of the Hawaiʻi Department of Health or the University of Hawaiʻi.

**Competing interests:** The authors have declared that no competing interests exist.

Population-level surveillance shows that this lack of activity is associated with an increased risk of noncommunicable diseases and premature mortality [3,4]. Native Hawaiians and other Pacific Islanders have been reported to have higher rates of physical inactivity than other racial/ethnic groups in the United States [5]. Historically, structural barriers, colonization, and discrimination have profoundly influenced the socioeconomic standing of these populations, resulting in poor health outcomes across their lifetime [6,7]. As is the case with many Indigenous populations globally, there is limited research on physical activity interventions relevant to these populations from a strengths-based, culturally responsive perspective [8]. There is also a need for community-based health promotion strategies that foster collaboration across different sectors [9]; culturally relevant physical activity may help to address multiple barriers through a single activity.

Physical activities are embedded within the fabric of social gatherings and traditional practices in Pacific Island communities. Leveraging these community-based strategies for Native Hawaiians and other Pacific Islanders, as well as other minority coastal communities, may effectively promote positive health behaviors and foster intergenerational health promotion and engagement. Prior to colonialization, ancestral Hawaiians held a profound connection between the mind, body, and the natural environment, which is reflected in their cultural practices that continue to this day. Culturally relevant physical activity, based on a population's norms, preferences, and customs, is thus one strategy with the potential to reduce the disease burden and health inequities associated with physical inactivity, promoting the collective well-being of the community [10]. For example, past studies have investigated the effects of practicing hula on chronic disease and observed a decreased risk of hypertension [11]. Despite the value of these culturally relevant activities, they are not typically measured in global, national, or population-level health surveys; therefore, the degrees of engagement in these activities often remain unreported [10].

## Fishing as a culturally based physical activity in the pacific

Living on islands, fishing is engrained in traditional Indigenous knowledge systems, with certain techniques offering unique physical activity opportunities [12]. It is believed that during the peak of the Hawaiian Kingdom, nearly every Hawaiian resident participated in some form of fishing [13]. Even after the annexation of the Hawaiian Kingdom and the subsequent transition from a cultural subsistence-based management system to a commercial economy, many traditional fishing practices, including spearfishing, are still in use today [13].

Spearfishing is a more active form of recreational fishing, embodying a diverse range of techniques and tools across various cultures worldwide, and involves the use of a specialized spear or harpoon to catch fish underwater [14]. There are different approaches to spearfishing. One method is free diving, which relies on breath-holding and diving without an additional breathing apparatus, demanding considerable skill in breath control and swimming. The use of snorkeling gear, or even SCUBA, can enable deeper dives and extended periods underwater without losing access to air. Reef walking, or the practice of traversing reef flats and shallow waters on foot while hunting for fish, is another method that requires keen observation and ecosystem knowledge.

Spearfishing serves as a culturally relevant practice that encourages a sense of community and positive social identity [13,14]. It also contributes to food security due to its high efficiency in catching desirable reef fish [15]. In the state of Hawaiʻi, research on fishing generally highlights its importance for food security and social cohesion. One study on the island of Hawaiʻi documented that 92% of fish caught remained within a 75-km radius of where they were caught, with half being kept and another third being given away. Additionally, a full fifth of all fish caught was used for cultural events such as birthdays, weddings, and funerals [16]. Despite its numerous health-promoting factors, such as social engagement, food

security, nature connectedness, and physical activity, it is insufficiently explored in public health research. There are sustainability concerns with spearfishing. It has gained popularity in recent decades, prompting concerns about the decline of marine species. As a result, the conservation community has called for restrictions on certain forms of spearfishing, such as SCUBA and night-time spearfishing. However, alternative fishery management approaches, such as implementing bag and size limits, have been suggested to help lessen the impact of spearfishing on marine ecosystems [17].

### Spearfishing and health outcomes

Most literature on spearfishing focuses on food security, marine conservation, and ecosystem impacts [14,15,18,19]. More health promotion research exists on components of the activity, including open-water swimming and diving (freediving and SCUBA). Depending on the individual in question, the CDC categorizes recreational swimming and diving as moderate to vigorous physical activity [20]. Open-water swimming may reduce stress, tension, symptoms of depression, fatigue, and memory problems while providing a sense of community and personal achievement to participants [21–23].

Free diving is often associated with spearfishing, and many of the benefits associated with open water swimming can be presumed to apply to free diving. While SCUBA spearfishing is a notable technique in spearfishing, it is banned in several countries, and in select marine areas around the Hawaiian Islands [24]. However, SCUBA diving has been shown to encourage social interaction, promote a meditative mindset, induce feelings of relaxation, and reduce symptoms of post-traumatic stress disorder, depression, and anxiety, especially among those with disabilities [25–27]. This is partially due to the accessibility of marine environments to individuals of all conditions, but also the relaxing effect the aquatic environment contributes to the body, improving movement and coordination [26–28].

The natural surroundings where spearfishing takes place also contribute to health benefits. In health promotion, there is growing interest in examining the physical and mental health benefits of spending time in outdoor settings [29,30]. Some studies have associated positive health outcomes with recreational activities in these areas, such as improvements to cardiovascular fitness [31]. Other studies have attributed time spent in outdoor blue spaces to improved cognitive and immune function, reduced mental distress, and improved overall life satisfaction [32–36].

### Study objective

This study estimates the prevalence of lifetime engagement with spearfishing in Hawai'i and associated sociodemographic and health factors from a health promotion perspective. The Hawaiian Islands provide an ideal setting for this work, given their central Pacific location and the diverse population with important cultural ties to spearfishing. Building on findings of higher lifetime engagement in culturally relevant activities such as hula and paddling among Native Hawaiians and Other Pacific Islanders [10], and considering the extensive history of fishing in the Hawaiian Islands [13], we hypothesize a high prevalence of spearfishing engagement, particularly among Native Hawaiian and Pacific Islander communities. Furthermore, we hypothesize that participation would vary based on sociodemographic and health factors, with younger individuals and those in better health being more likely to participate.

## Methods

### Study design and population

The study employed a cross-sectional design using data from the 2019 and 2020 waves of the Behavioral Risk Factor Surveillance System (BRFSS) survey. The BRFSS is an annual, national

telephone survey program sponsored by the U.S. Centers for Disease Control and Prevention (CDC) that collects data about demographic characteristics, current health behaviors, and health status across the U.S. [37]. It is the largest continuously conducted health survey performed in the country and annually conducts interviews with adults aged 18 and older, covering all 50 states, the District of Columbia, Puerto Rico, and Guam. The primary survey consists of a core questionnaire addressing essential public health issues, optional additional models, and any state-specific questions [37]. States have the option to include some questions of their own design to allow for flexibility in surveillance based on state-level health priorities. These are administered at the end of the survey. In 2019 and 2020, one question about spearfishing was added to the Hawai'i BRFSS (H-BRFSS). Our sample was obtained from the 2019-2020 H-BRFSS surveys.

## Variables

**Outcome.** Our outcome measure was self-reported lifetime engagement in spearfishing, ascertained by the following question: *"Over your lifetime, how much have you engaged with spearfishing?"* with the response choices being: never, almost never, sometimes, often, and very often. Lifetime engagement in spearfishing was dichotomized to compare those who reported spearfishing "sometimes," "often" or "very often" during their lifetimes with those who reported participating 'never' or 'almost never'.

**Correlates.** In this study, we considered various demographic, socioeconomic, behavioral, and health variables: sex assigned at birth (female or male), age group (18–24, 25–34, 35–44, 45–54, 55–64, or ≥65 y), race/ethnicity, level of education (high school diploma or less; some college; college graduate), federal poverty level (0-130%, 131-185%, 185% and greater), island of residence (O'ahu, Hawai'i Island, Kaua'i, Maui, Moloka'i, or Lāna'i, and marital status (married or partnered, or single). The classification of "race and ethnicity" adhered to the standards set by the Hawai'i State Department of Health, encompassing diverse groups relevant to the state's demographics: White, Native Hawaiian, Other Pacific Islander, Japanese, Chinese, Filipino, Other Asian, American Indian or Alaskan Native, Black, or other group [38].

Health behavior variables were also considered. These included whether or not the participant had smoked 100 cigarettes or more in their lifetime, met US physical activity guidelines of 150 minutes per week or 75 minutes of vigorous equivalents per week (yes/no) (only assessed in 2019), and visited a doctor in the last 12 months [39]. A number of self-reported health status variables were also examined. These included body mass index (BMI; normal, underweight, overweight, obese) calculated according to a participant's reported height and weight, self-rated health (excellent/very good, good, or fair/poor), and self-reported dichotomous chronic diseases: depressive disorder, diabetes, hypertension/high blood pressure (only asked in year 2019), heart disease, arthritis, and asthma. Finally, two measures of functional limitation were included: difficulty walking or climbing stairs, and difficulty dressing or bathing. For those questions only asked in 2019 (physical activity guidelines and hypertension), we coded the missing responses for 2020 as unknown. This allows for assessment of differences between those with and without responses for those variables. S1 provides more details on the variable coding and variable labels.

## Sample

A total of N = 15,437 respondents completed the 2019 and 2020 (7,683 in 2019 and 7,754 in 2020) H-BRFSS. There were 1948 (12.7%) missing responses for the outcome measure of spearfishing. There were 1973 missing responses for poverty level and 713 for BMI. After listwise deletion of participants with missing observations (5.6% of 13489) for other variables,

the sample size was 12,373. Because about 10% of participants had missing values for poverty level, they were retained, and results are presented across poverty level categories and by "Don't know/refused.".

## Statistical analyses

S2 provides the dataset used for the analyses. Weighted analyses were employed to account for the complex sampling design, with equal weighting for 2019 and 2020. Descriptive statistics, including both the unweighted and weighted percentages, were used to describe the participant characteristics. For each correlate examined, we calculated the crude and adjusted prevalence ratios (PRs), and their corresponding 95% confidence intervals (95%CI), for engagement with spearfishing. A prevalence ratio (PR) is commonly used to assess the strength of the association between an independent variable and outcome of interest [40]. It compares the prevalence of an outcome in those who have a certain characteristic to those who do not. This analysis was conducted using a weighted Poisson regression with a quasi-likelihood estimating method. The choice of Poisson regression over logistic regression was made due to the potential for odds ratios to overestimate the prevalence ratio, particularly when prevalence is close to or greater than 10% [40].

Three multivariate Poisson regression models were constructed to examine the adjusted PR for each correlate while simultaneously considering other variables. The first model adjusted for demographic characteristics: sex, age group, race/ethnicity, level of education, federal poverty level, island of residence, and marital status. The second model adjusted for all health behaviors: smoking at least 100 cigarettes in a lifetime, meeting physical activity guidelines, and doctor's visit in the past 12 months. The third model additionally adjusted for health status variables, including BMI, self-rated health, and chronic disease and functional limitation variables. This modeling approach provides insights into how other correlates relate to each other when assessing changes in the association between correlates and the spearfishing outcome. Crude models present unadjusted associations, showing the relationship between a single characteristic and outcome variable without accounting for other variables that could influence the results. Adjusted models incorporate additional variables, typically to control for confounding effects, providing a clearer understanding of the independent association between a characteristic and outcome. Results were deemed statistically significant if the p-value was 0.05 or less. These analyses were repeated using an outcome measure of lifetime spearfishing often/very often, rather than sometimes/often/very often assess associations with individuals who consider themselves frequent spearfishers. The results of this analysis are presented in the supplementary files.

We conducted the analysis of demographic characteristics using SAS 9.4 (SAS Institute., Cary, NC), and R version 4.05 was used for weighted Poisson regression models.

## Ethical approval

This study utilizes publicly available BRFSS data. The analyses were conducted on de-identified, publicly available datasets that do not constitute human subjects research as defined by 45 CFR 46.102 of the United States of America.

## Results

### Population characteristics

An estimated 25.5% of the adult population of Hawaiʻi engaged in spearfishing sometimes, often, or very often during their lifetimes. The weighted population size of those engaging with spearfishing during their lifetimes was 226,814 people, with 89,750 estimated to have done so often or very often (Table 1). Detailed population characteristics of the sample are provided in Table 1.

**Table 1. Demographic and Health Characteristics of Participants in the 2019 and 2020 H-BRFSS.**

| | Sample size | Sample Proportion (%) | Weighted Population size | Weighted Proportion (%) |
|---|---|---|---|---|
| **Total Sample** | 12,737 | 100 | 890,850 | 100 |
| **Sex** | | | | |
| Female | 6,702 | 52.6 | 441,740 | 49.6 |
| Male | 6,035 | 47.4 | 449,110 | 50.4 |
| **Age group (years)** | | | | |
| 18-24 | 800 | 6.3 | 89,713 | 10.1 |
| 25-34 | 1,423 | 11.2 | 145,191 | 16.3 |
| 35-44 | 1,714 | 13.5 | 146,427 | 16.4 |
| 45-54 | 1,791 | 14.1 | 131,181 | 14.7 |
| 55-64 | 2,440 | 19.2 | 149,984 | 16.8 |
| 65+ | 4,569 | 35.9 | 228,355 | 25.6 |
| **Race/Ethnicity** | | | | |
| White | 4,643 | 36.5 | 227,459 | 25.5 |
| Native Hawaiian | 2,358 | 18.5 | 160,387 | 18.0 |
| Other Pacific Islander | 517 | 4.1 | 43,407 | 4.9 |
| Japanese | 1,935 | 15.2 | 161,454 | 18.1 |
| Chinese | 580 | 4.6 | 56,572 | 6.4 |
| Filipino | 1,414 | 11.1 | 132,118 | 14.8 |
| Other Asian | 554 | 4.3 | 51,693 | 5.8 |
| American Indian or Alaskan Native | 243 | 1.9 | 13,401 | 1.5 |
| Black | 196 | 1.5 | 22,029 | 2.5 |
| Other | 297 | 2.3 | 22,329 | 2.5 |
| **Education Level** | | | | |
| College Graduate | 5,531 | 43.4 | 277,166 | 31.1 |
| Some College | 3,633 | 28.5 | 292,788 | 32.9 |
| High School Diploma or Less | 3,573 | 28.1 | 320,897 | 36.0 |
| **Federal Poverty Level** | | | | |
| 0-130% | 2,839 | 22.3 | 205,918 | 23.1 |
| 131-185% | 1,318 | 10.3 | 101,163 | 11.4 |
| 186+% | 7,374 | 57.9 | 493,395 | 55.4 |
| Don't know/refused | 1,206 | 9.5 | 90,375 | 10.1 |
| **Island of Residence** | | | | |
| Oʻahu | 6,302 | 49.5 | 607,917 | 68.2 |
| Hawaiʻi Island | 2,650 | 20.8 | 131,138 | 14.7 |
| Kauai | 1,510 | 11.9 | 46,920 | 5.3 |
| Maui | 1,973 | 15.5 | 97,151 | 10.9 |
| Molokaʻi | 205 | 1.6 | 5,832 | 0.7 |
| Lānaʻi | 97 | 0.8 | 1,892 | 0.2 |
| **Marital Status** | | | | |
| Married or partnered | 6,968 | 54.7 | 508,579 | 57.1 |
| Single | 5,769 | 45.3 | 382,271 | 42.9 |
| **Ever smoked 100 cigarettes** | | | | |
| No | 7,628 | 59.9 | 555,717 | 62.4 |
| Yes | 5,109 | 40.1 | 335,134 | 37.6 |
| **Met Physical Activity Guidelines** | | | | |

*(Continued)*

**Table 1.** (Continued)

| | Sample size | Sample Proportion (%) | Weighted Population size | Weighted Proportion (%) |
|---|---|---|---|---|
| Did not meet | 2,414 | 19.0 | 185,566 | 20.8 |
| Met | 3,657 | 28.7 | 240,074 | 26.9 |
| 2020/Don't know/Refused | 6,666 | 52.3 | 465,210 | 52.2 |
| **Visited a doctor in the last 12 months** | | | | |
| No | 2,363 | 18.6 | 171,968 | 19.3 |
| Yes | 10,374 | 81.4 | 718,883 | 80.7 |
| **BMI Categories** | | | | |
| Normal | 5,043 | 39.6 | 344,053 | 38.6 |
| Overweight | 4,223 | 33.2 | 297,180 | 33.4 |
| Obese | 3,111 | 24.4 | 222,778 | 25.0 |
| Underweight | 360 | 2.8 | 26,841 | 3.0 |
| **Self-Rated Health** | | | | |
| Excellent/very good | 6,845 | 53.7 | 477,321 | 53.6 |
| Good | 4,079 | 32.0 | 294,884 | 33.1 |
| Fair/poor | 1,813 | 14.2 | 118,646 | 13.3 |
| **Depressive Disorder** | | | | |
| No | 10,905 | 85.6 | 776,056 | 87.1 |
| Yes | 1,832 | 14.4 | 114,794 | 12.9 |
| **Diabetes** | | | | |
| No | 11,251 | 88.3 | 790,056 | 88.7 |
| Yes | 1,486 | 11.7 | 100,794 | 11.3 |
| **High Blood Pressure** | | | | |
| No | 4,043 | 31.7 | 300,526 | 33.7 |
| Yes | 2,219 | 17.4 | 140,740 | 15.8 |
| 2020/Don't know/refused | 6,475 | 50.8 | 449,584 | 50.5 |
| **Heart Disease** | | | | |
| No | 12,260 | 96.3 | 864,336 | 97.0 |
| Yes | 477 | 3.7 | 26,515 | 3.0 |
| **Arthritis** | | | | |
| No | 9,372 | 73.6 | 698,703 | 78.4 |
| Yes | 3,365 | 26.4 | 192,148 | 21.6 |
| **Asthma** | | | | |
| No | 10,862 | 85.3 | 751,937 | 84.4 |
| Yes | 1,875 | 14.7 | 138,913 | 15.6 |
| **Difficulty Walking or Climbing up stairs** | | | | |
| No | 11,221 | 88.1 | 800,545 | 89.9 |
| Yes | 1,516 | 11.9 | 90,306 | 10.1 |
| **Difficulty Dressing or bathing** | | | | |
| No | 12,395 | 97.3 | 870,279 | 97.7 |
| Yes | 342 | 2.7 | 20,572 | 2.3 |
| **Spearfishing** | | | | |
| 1. Never | 8,627 | 67.7 | 590,878 | 66.3 |
| 2. Almost never | 1,031 | 8.1 | 73,158 | 8.2 |
| 3. Sometimes | 1,878 | 14.7 | 137,064 | 15.4 |
| 4. Often | 691 | 5.4 | 51,484 | 5.8 |
| 5. Very often | 510 | 4.0 | 38,266 | 4.3 |

The crude and adjusted PRs for the association of each correlate with lifetime spearfishing engagement were calculated across all models (Table 2). Men reported much higher lifetime engagement in spearfishing (40.5%) compared to women (10.2%; PR: 3.98, 95%CI: 3.57-4.43). The association was robust across all models. Lifetime spearfishing engagement was relatively consistent across age groups, varying from 29.3% (95%CI: 26.5-32.2%) among those 35 to 44 years to 21.6% (95%CI: 19.9-23.4%) among those 65 and older. When accounting for other covariates in models 1-3, there were no statistically significant differences between the different age categories when compared to the youngest group (18-24).

The prevalence of lifetime spearfishing engagement was highest among Native Hawaiians (42.6%, 95%CI: 40.0-45.2), other Pacific Islanders (36.4%, 95%CI: 31.3-41.8), American Indian or Alaskan Native (32.3%, 95%CI: 24.8-41.0) and Japanese (26.2%, 95%CI: 23.8-28.7). With the exception of Chinese and Black race/ethnicity categories, all other groups were significantly more likely to spearfish in their lifetimes compared to Whites.

The prevalence of spearfishing varied by education level; 30.0% (95%CI: 28.0-32.0%) of those with a high school diploma or less reported spearfishing engagement, compared to 19.8% of college graduates (95%CI: 18.5-21.2%). In Model 1, which included education and poverty level together, the PR for those with a high school diploma or less, compared to college graduates, attenuated (1.51, 95%CI: 1.38-1.67 unadjusted to 1.19, 95%CI: 1.07-1.31 in Model 1), as did the PR for some college (1.31, 95%CI: 1.19-1.45 unadjusted to 1.17, 95%CI: 1.07-1.29). The PRs remained relatively similar in subsequent models, which included health behaviors and conditions.

In the unadjusted analyses, those at 130% of the poverty line or below were significantly more likely to have engaged in spearfishing sometimes/often/very often in their lifetimes compared to those with higher incomes. Once other covariates were included in the Model 1, the association was no longer statistically significant and remained relatively similar in the subsequent 2 models. There was also a strong variation in lifetime engagement according to the island of residence. On the most populous island of O'ahu, the prevalence of spearfishing was 22.8%, compared to the rural islands of Moloka'i (43.1%) and Lāna'i (50.8%). The associations between island and spearfishing varied little across models, and the prevalence of spearfishing engagement was significantly higher on all other islands compared to O'ahu.

Those who were married or partnered were more likely to report spearfishing engagement compared to those who were single. Adjustment for demographic factors accentuated the association when compared to the unadjusted PR. Further statistical adjustment had little to no effect on the estimates with a PR of 0.89 (95%CI: 0.82-0.96) across all other model

## Prevalence and associations of spearfishing engagement by health behaviors and status

There were associations between lifetime spearfishing engagement, smoking, and meeting physical activity guidelines. There was a borderline association with having visited the doctor in the past year. The prevalence of lifetime spearfishing engagement was significantly higher among those reporting smoking 100 cigarettes or more during their lifetimes compared to those who did not (PR: 1.43, 95%CI: 1.38-1.62). The PR became smaller after statistical adjustment (PR: 1.25 in both Model 2 and 3), but remained statistically significant (95%CI: 1.16-1.35). Those reporting spearfishing sometimes/often/very often during their lifetimes were significantly more likely to meet physical activity guidelines (PR: 1.44 95%CI: 1.27-1.64) with the PRs largely unchanged across models. The direction of association switched for having seen the doctor in the past year after other health behaviors and health status indicators were included in the models (PR: 0.92, $p$-value 0.10 unadjusted to 1.10, $p$-value 0.05 in Model 3).

**Table 2. Prevalence of Spearfishing (Sometimes, Often, and Very Often), by Demographic and Health Characteristics, Among Participants in the 2019 and 2020 H-BRFSS.**

| | Prevalence (%) and 95% CI | cPR (95% CI) | p-value | Model 1 aPR (95% CI) | p-value | Model 2 aPR (95% CI) | p-value | Model 3 aPR (95% CI) | p-value |
|---|---|---|---|---|---|---|---|---|---|
| **Total Sample** | 25.5 (24.4, 26.5) | | | | | | | | |
| **Sex** | | | | | | | | | |
| Female | 10.2 (9.2, 11.2) | Ref | | | | | | | |
| Male | 40.5 (38.8, 42.2) | 3.98 (3.57, 4.43) | <0.0001 | 4.04 (3.63, 4.49) | <0.0001 | 3.97 (3.57, 4.41) | <0.0001 | 3.93 (3.53, 4.37) | <0.0001 |
| **Age group (years)** | | | | | | | | | |
| 18-24 | 26.4 (22.8, 30.4) | Ref | | | | | | | |
| 25-34 | 26.9 (24.2, 29.8) | 1.02 (0.85, 1.22) | 0.8331 | 1.03 (0.88, 1.21) | 0.7205 | 1.00 (0.85, 1.17) | 0.9792 | 0.98 (0.84, 1.16) | 0.8421 |
| 25-44 | 29.3 (26.5, 32.2) | 1.11 (0.93, 1.32) | 0.2522 | 1.17 (1.00, 1.38) | 0.0499 | 1.11 (0.94, 1.30) | 0.2182 | 1.08 (0.92, 1.27) | 0.3488 |
| 45-54 | 25.1 (22.7, 27.7) | 0.95 (0.80, 1.13) | 0.5711 | 1.00 (0.85, 1.18) | 0.9981 | 0.95 (0.81, 1.11) | 0.5154 | 0.91 (0.77, 1.07) | 0.2630 |
| 55-64 | 25.9 (23.6, 28.3) | 0.98 (0.83, 1.16) | 0.8075 | 1.10 (0.94, 1.28) | 0.2457 | 1.03 (0.88, 1.20) | 0.7531 | 0.98 (0.84, 1.16) | 0.8511 |
| 65+ | 21.6 (19.9, 23.4) | 0.82 (0.69, 0.96) | 0.0169 | 1.01 (0.87, 1.18) | 0.8614 | 0.91 (0.78, 1.06) | 0.2148 | 0.86 (0.73, 1.02) | 0.0750 |
| **Race/Ethnicity** | | | | | | | | | |
| White | 17.5 (15.9, 19.1) | Ref | | | | | | | |
| Native Hawaiian | 42.6 (40.0, 45.2) | 2.44 (2.18, 2.73) | <0.0001 | 2.59 (2.32, 2.90) | <0.0001 | 2.62 (2.34, 2.92) | <0.0001 | 2.55 (2.27, 2.86) | <0.0001 |
| Other Pacific Islander | 36.4 (31.3, 41.8) | 2.08 (1.75, 2.47) | <0.0001 | 2.26 (1.91, 2.66) | <0.0001 | 2.28 (1.93, 2.69) | <0.0001 | 2.21 (1.87, 2.61) | <0.0001 |
| Japanese | 26.2 (23.8, 28.7) | 1.50 (1.31, 1.71) | <0.0001 | 1.77 (1.56, 2.00) | <0.0001 | 1.81 (1.61, 2.05) | <0.0001 | 1.80 (1.59, 2.04) | <0.0001 |
| Chinese | 15.7 (12.4, 19.5) | 0.90 (0.70, 1.14) | 0.3796 | 1.08 (0.85, 1.37) | 0.5447 | 1.14 (0.90, 1.44) | 0.2901 | 1.14 (0.90, 1.44) | 0.2837 |
| Filipino | 21.2 (18.7, 23.9) | 1.21 (1.04, 1.41) | 0.0143 | 1.44 (1.24, 1.67) | <0.0001 | 1.52 (1.31, 1.76) | <0.0001 | 1.50 (1.29, 1.74) | <0.0001 |
| Other Asian | 21.7 (17.4, 26.8) | 1.24 (0.98, 1.57) | 0.0672 | 1.49 (1.18, 1.88) | 0.0007 | 1.52 (1.21, 1.92) | 0.0004 | 1.53 (1.21, 1.92) | 0.0003 |
| American Indian or Alaskan Native | 32.3 (24.8, 41.0) | 1.85 (1.42, 2.42) | <0.0001 | 1.73 (1.35, 2.20) | <0.0001 | 1.73 (1.35, 2.21) | <0.0001 | 1.70 (1.33, 2.18) | <0.0001 |
| Black | 13.6 (8.1, 21.9) | 0.78 (0.47, 1.29) | 0.3289 | 0.82 (0.51, 1.33) | 0.4175 | 0.83 (0.51, 1.34) | 0.4475 | 0.81 (0.51, 1.28) | 0.3640 |
| Other | 24.2 (17.2, 32.8) | 1.38 (0.99, 1.93) | 0.0573 | 1.32 (0.97, 1.80) | 0.0791 | 1.35 (0.99, 1.83) | 0.0578 | 1.34 (0.99, 1.82) | 0.0579 |
| **Education Level** | | | | | | | | | |
| College Graduate | 19.8 (18.5, 21.2) | Ref | | | | | | | |
| Some College | 25.9 (24.1, 27.8) | 1.31 (1.19, 1.45) | <0.0001 | 1.17 (1.07, 1.29) | 0.0009 | 1.15 (1.05, 1.26) | 0.0040 | 1.15 (1.05, 1.26) | 0.0037 |
| High School Diploma or Less | 30.0 (28.0, 32.0) | 1.51 (1.38, 1.67) | <0.0001 | 1.19 (1.07, 1.31) | 0.0007 | 1.15 (1.04, 1.27) | 0.0059 | 1.14 (1.03, 1.26) | 0.0094 |
| **Federal Poverty Level** | | | | | | | | | |
| 0-130% | 28.5 (26.3, 30.7) | Ref | | | | | | | |
| 131-185% | 24.5 (21.4, 28.0) | 0.86 (0.74, 1.01) | 0.0586 | 0.90 (0.78, 1.04) | 0.1718 | 0.92 (0.80, 1.06) | 0.2391 | 0.91 (0.79, 1.05) | 0.2169 |
| 186+% | 25.2 (23.9, 26.6) | 0.89 (0.81, 0.97) | 0.0111 | 0.98 (0.89, 1.07) | 0.6329 | 0.97 (0.89, 1.07) | 0.5621 | 0.97 (0.88, 1.07) | 0.5395 |
| Don't Know/Refused | 21.1 (17.9, 24.5) | 0.74 (0.62, 0.88) | 0.0007 | 0.82 (0.70, 0.96) | 0.0154 | 0.83 (0.71, 0.97) | 0.0211 | 0.83 (0.71, 0.97) | 0.0192 |
| **Island of Residence** | | | | | | | | | |
| O'ahu | 22.8 (21.5, 24.2) | Ref | | | | | | | |
| Hawai'i Island | 32.1 (29.7, 34.6) | 1.41 (1.28, 1.55) | <0.0001 | 1.37 (1.25, 1.50) | <0.0001 | 1.37 (1.25, 1.50) | <0.0001 | 1.37 (1.26, 1.50) | <0.0001 |
| Kauai | 31.5 (28.2, 35.0) | 1.38 (1.22, 1.56) | <0.0001 | 1.48 (1.32, 1.65) | <0.0001 | 1.47 (1.32, 1.64) | <0.0001 | 1.48 (1.33, 1.66) | <0.0001 |
| Maui | 28.4 (25.9, 31.0) | 1.24 (1.12, 1.38) | <0.0001 | 1.29 (1.17, 1.42) | <0.0001 | 1.28 (1.16, 1.40) | <0.0001 | 1.28 (1.16, 1.40) | <0.0001 |
| Moloka'i | 43.1 (32.0, 54.9) | 1.89 (1.43, 2.48) | <0.0001 | 1.68 (1.34, 2.10) | <0.0001 | 1.61 (1.30, 2.00) | <0.0001 | 1.63 (1.31, 2.02) | <0.0001 |
| Lāna'i | 50.8 (35.6, 65.8) | 2.22 (1.63, 3.02) | <0.0001 | 2.06 (1.50, 2.84) | <0.0001 | 1.97 (1.42, 2.72) | <0.0001 | 1.97 (1.44, 2.70) | <0.0001 |
| **Marital Status** | | | | | | | | | |
| Married or Partnered | 26.3 (25.0, 27.8) | Ref | | | | | | | |

*(Continued)*

**Table 2.** (Continued)

| | Prevalence (%) and 95% CI | cPR (95% CI) | p-value | Model 1 | | Model 2 | | Model 3 | |
|---|---|---|---|---|---|---|---|---|---|
| | | | | aPR (95% CI) | p-value | aPR (95% CI) | p-value | aPR (95% CI) | p-value |
| Single | 24.3 (22.8, 25.9) | 0.92 (0.85, 1.00) | 0.0555 | 0.89 (0.83, 0.97) | 0.0060 | 0.89 (0.82, 0.96) | 0.0035 | 0.89 (0.82, 0.96) | 0.0042 |
| **Ever smoked 100 cigarettes** | | | | | | | | | |
| No | 21.5 (20.2, 22.8) | Ref | | | | | | | |
| Yes | 32.1 (30.3, 33.9) | 1.49 (1.38, 1.62) | <0.0001 | | | 1.25 (1.16, 1.35) | <0.0001 | 1.25 (1.16, 1.35) | <0.0001 |
| **Met Physical Activity Guidelines** | | | | | | | | | |
| Did not meet | 19.6 (17.6, 21.7) | Ref | | | | | | | |
| Met | 28.2 (26.2, 30.3) | 1.44 (1.27, 1.64) | <0.0001 | | | 1.43 (1.28, 1.61) | <0.0001 | 1.45 (1.29, 1.63) | <0.0001 |
| 2020/Don't know/refused | 26.4 (25.0, 27.9) | 1.35 (1.20, 1.52) | <0.0001 | | | 1.33 (1.20, 1.48) | <0.0001 | 1.23 (0.95, 1.61) | 0.1174 |
| **Visited a Doctor in the Last 12 Months** | | | | | | | | | |
| No | 27.2 (24.9, 29.6) | Ref | | | | | | | |
| Yes | 25.0 (23.9, 26.2) | 0.92 (0.84, 1.02) | 0.0979 | | | 1.11 (1.01, 1.23) | 0.0262 | 1.10 (1.00, 1.21) | 0.0479 |
| **BMI Categories** | | | | | | | | | |
| Normal | 19.9 (18.4, 21.5) | Ref | | | | | | | |
| Overweight | 27.9 (26.1, 29.8) | 1.41 (1.27, 1.56) | <0.0001 | | | | | 1.05 (0.95, 1.15) | 0.3574 |
| Obese | 32.2 (30.0, 34.5) | 1.62 (1.46, 1.80) | <0.0001 | | | | | 1.10 (0.99, 1.22) | 0.0806 |
| Underweight | 14.0 (9.7, 19.8) | 0.71 (0.49, 1.02) | 0.0607 | | | | | 0.88 (0.62, 1.26) | 0.4838 |
| **Self-Rated Health** | | | | | | | | | |
| Excellent/Very Good | 24.1 (22.8, 25.5) | Ref | | | | | | | |
| Good | 25.9 (24.1, 27.8) | 1.07 (0.98, 1.18) | 0.1321 | | | | | 0.92 (0.84, 1.00) | 0.0475 |
| Fair/Poor | 29.7 (27.0, 32.6) | 1.23 (1.10, 1.38) | 0.0002 | | | | | 0.99 (0.88, 1.11) | 0.8570 |
| **Depressive Disorder** | | | | | | | | | |
| No | 25.9 (24.8, 27.0) | Ref | | | | | | | |
| Yes | 22.4 (19.9, 25.1) | 0.86 (0.76, 0.98) | 0.0211 | | | | | 0.93 (0.83, 1.05) | 0.2311 |
| **Diabetes** | | | | | | | | | |
| No | 24.6 (23.5, 25.7) | Ref | | | | | | | |
| Yes | 32.1 (28.9, 35.5) | 1.30 (1.17, 1.46) | <0.0001 | | | | | 1.10 (0.98, 1.23) | 0.1010 |
| **High Blood Pressure** | | | | | | | | | |
| No | 23.4 (21.7, 25.2) | Ref | | | | | | | |
| Yes | 27.0 (24.5, 29.6) | 1.15 (1.02, 1.30) | 0.0223 | | | | | 0.96 (0.86, 1.08) | 0.4786 |
| 2020/Don't Know/Refused | 26.4 (24.9, 27.9) | 1.13 (1.02, 1.24) | 0.0132 | | | | | 1.07 (0.83, 1.38) | 0.6150 |
| **Heart Disease** | | | | | | | | | |
| No | 25.2 (24.2, 26.3) | Ref | | | | | | | |
| Yes | 34.1 (28.0, 40.7) | 1.35 (1.12, 1.64) | 0.0021 | | | | | 1.09 (0.92, 1.28) | 0.3175 |
| **Arthritis** | | | | | | | | | |
| No | 25.5 (24.4, 26.7) | Ref | | | | | | | |
| Yes | 25.2 (23.2, 27.4) | 0.99 (0.90, 1.09) | 0.8114 | | | | | 1.02 (0.93, 1.11) | 0.6733 |
| **Asthma** | | | | | | | | | |
| No | 25.1 (24.0, 26.2) | Ref | | | | | | | |
| Yes | 27.5 (24.7, 30.4) | 1.09 (0.98, 1.23) | 0.1154 | | | | | 1.08 (0.98, 1.20) | 0.1239 |

*(Continued)*

**Table 2.** (Continued)

| | Prevalence (%) and 95% CI | cPR (95% CI) | p-value | Model 1 | | Model 2 | | Model 3 | |
|---|---|---|---|---|---|---|---|---|---|
| | | | | aPR (95% CI) | p-value | aPR (95% CI) | p-value | aPR (95% CI) | p-value |
| **Difficulty Walking or Climbing up stairs** | | | | | | | | | |
| No | 25.3 (24.3, 26.4) | Ref | | | | | | | |
| Yes | 26.6 (23.5, 29.9) | 1.05 (0.92, 1.19) | 0.4709 | | | | | 1.13 (0.99, 1.30) | 0.0722 |
| **Difficulty Dressing or bathing** | | | | | | | | | |
| No | 25.5 (24.4, 26.5) | Ref | | | | | | | |
| Yes | 25.4 (19.9, 31.8) | 1.00 (0.78, 1.27) | 0.9790 | | | | | 0.79 (0.63, 0.99) | 0.0437 |

There was a higher prevalence of spearfishing engagement among those who were overweight or obese in the unadjusted model. However, the association became non-significant for overweight (PR: 1.05, 95%CI: 0.95-1.15) and marginally significant for obese individuals (PR:1.10, 95%CI: 0.99-1.22) in Model 3, which statistically adjusted for demographic characteristics, health behaviors, and other health status indicators. Unadjusted, those reporting fair and poor health were more likely to report lifetime spearfishing engagement compared to those in excellent or very good health. This association disappeared once demographic and health behavior variables were included in the model.

There were also associations between spearfishing and diabetes and heart disease in the unadjusted models. These associations were lost after adjusting for sociodemographic characteristics and health behaviors. Other than BMI, the only other health status variable that was statistically significantly associated with spearfishing was difficulty dressing or bathing; those with this functional limitation were less likely to have engaged in spearfishing during their lifetimes than those without it. There was a marginal association between difficulty walking or climbing stairs and spearfishing engagement, with an increase in PR from 1.05 (95%CI: 0.92-1.19) in the Unadjusted Model to 1.13 (95%CI: 0.99-1.30) in Model 3.

We also conducted an additional analysis focusing on individuals who reported frequent spearfishing (see S1 Table), which represented 10.1% of Hawaiʻi's adult population. Results were mostly similar. The strength of association intensified between Native Hawaiian and Other Pacific Islander ethnicities and lifetime spearfishing when compared to Whites, as did the associations with living on rural islands, and lower educational attainment. In addition to the association already observed between obesity and lifetime spearfishing engagment, those in the overweight category were also significantly more likely to report spearfishing often or very often. Individuals who reported difficulty walking or climbing stairs in the previous analyses were slightly more likely to engage in spearfishing (PR 1.13, 95%CI 0.99-1.30). This association became statistically significant when considering only those who reported doing it often or very often.

## Discussion

Based on the results of a prior study that documented elevated lifetime engagement in other culturally relevant activities in Hawaiʻi—hula and outrigger paddling—using the H-BRFSS [10], we anticipated a similar high engagement in spearfishing. The results of this study support this hypothesis, with 1 in 4 Hawaiʻi residents reporting engaging in spearfishing sometimes, often, or very often during their lifetime. The decision to include "sometimes" in the definition of spearfishing engagement reflects the potential health benefits of moderate

physical activity. According to the World Health Organization (WHO) global guidelines on physical activity and sedentary behavior, 'any amount of physical activity is better than none' [41]. Including "sometimes" in the engagement group thus acknowledges that moderate involvement in the activity may contribute to physical activity and cultural connection. Similar results to this paper were observed by Sentell et al. (2023) for hula and outrigger paddling, where it was estimated that 25% and 20% of the Hawaiʻi population engaged in these activities, respectively. Also similar to the Sentell et al. paper (2023), outrigger paddling was more common among men. In this work, 40% of men reported lifetime spearfishing engagement compared to 10% of women [10].

In accordance with our hypothesis, we also found that demographic and health-related correlates were associated with lifetime spearfishing engagement. Besides sex, the most notable associations were for race/ethnicity and island of residence. Two of five Native Hawaiians, and about one in three other Pacific Islanders, as well as American Indian or Alaskan Native, reported engaging in spearfishing during their lifetime. For Native Hawaiians and other Pacific Islanders, statistical adjustment for sociodemographic and health factors increased the magnitude of the association with spearfishing engagement. Native Hawaiians and other Pacific Islanders are also among the fastest-growing racial/ethnic groups in the US, further highlighting the importance of targeted, culturally relevant health interventions for them [42]. Aside from those who identified as Chinese or Black, all other racial demographic groups were statistically more likely to report spearfishing engagement compared to Whites.

There was very high lifetime engagement in spearfishing on the rural islands of Molokaʻi and Lānaʻi, which both have small populations (<10,000) [43,44]. Rural communities face a significant burden of chronic diseases, including elevated rates of hypertension and diabetes [45]. Identifying culturally relevant activities suitable to rural communities may assist in the development and application of health promotion activities directed at addressing chronic disease. The rural islands of Hawaiʻi also have significant populations of Native Hawaiians and other Pacific Islanders [46]. Thus, our findings are particularly relevant for initiatives aiming to promote physical activity among these groups, which experience some of the highest rates of chronic disease and associated adverse health outcomes in the country [47]. Native Hawaiians and other Pacific Islanders are 2.5 times more likely to be diagnosed with diabetes, 10% more likely to be diagnosed with coronary heart disease, 80% more likely to be obese, and four times as likely to have a stroke compared to non-Hispanic whites [48].

The health disparities experienced by Native Hawaiians and other Pacific Islanders are rooted in structural inequalities, colonization, and socioeconomic factors [47]. The elevated rates of obesity observed among reported spearfishers in our unadjusted models thus highlight broader population-level patterns and systemic challenges likely unrelated to the activity of spearfishing itself [47]. Furthermore, research quantifying the relationship between physical activity and adverse health outcomes among Native Hawaiians and Other Pacific Islanders found that only half of the surveyed participants reported meeting the US Department of Health and Human Services' physical activity guidelines of 150 minutes or 75 minutes of vigorous equivalent minutes per week, and a third reported no activity at all [39].

The prevalence of lifetime spearfishing engagement showed minimal variation across age groups, with only a five percent difference between the highest reported engagement (26.9% among individuals aged 25-34 years) and the lowest (21.6% among those > 65 years). The findings suggest that spearfishing may serve as an accessible activity for people of various ages at some point in their lifetime. The idea of spearfishing as an accessible activity is further supported by its associations with difficulty walking or climbing stairs. Beyond equipment costs, participation in spearfishing is widely accessible to those who can swim and access shorelines.

Past studies have looked at open-water swimming and diving interventions as a way to foster physical activity for those with disabilities [22,26,28,49].

Select case reports have indicated that open-water swimming and exposure to salty sea air could improve lung capacity and function, particularly for those with chronic obstructive pulmonary disease (COPD), asthma, and related diseases/disabilities [21,49]. Thus, learning to breathe correctly due to diving regularly may positively affect respiratory health. Additionally, the breathing techniques used by free divers have been implemented as a method in pulmonary rehabilitation and aid in the reduction of breathing difficulty in COPD patients, improving their exercise capacity and enhancing overall quality of life [50]. This suggests that diving and its' respective techniques could reduce the burden of certain chronic diseases among participants.

There were also significant associations between lifetime spearfishing engagement and education and poverty level. A robust association existed between having a high school diploma or less and spearfishing during one's life. For income, there was initially a moderate association between being above the state poverty level and spearfishing, but once education and other socio-demographic variables were included in the models, this association disappeared. While spearfishing equipment can be expensive [51] the association with poverty level in the unadjusted model was due to a number of other factors; those living on the rural islands, Native Hawaiians and other Pacific Islanders, and those with low educations tend have lower household incomes [48,52,53].

Finally, those who reported being single were moderately less likely to report spearfishing engagement than those who were married/partnered. Spearfishing may be important for maintaining household food security. A recent statewide analysis of fishing for household consumption and food insecurity found that food insecurity was higher in households that fished for food frequently than those that did not, suggesting that fishing may be an important coping strategy for food insecurity [54]. Fishing, generally, in Hawai'i is an important contributor to both household and community food security. Moreover, fishing plays an important role in social cohesion, and the provisioning of food at important cultural events helps to maintain social and kinship ties among community members and families [16].

## Limitations

There are limitations in this study. Due to the geographic focus on Hawai'i, our study primarily examined marine spearfishing. However, we acknowledge the existence of other spearfishing forms in freshwater ecosystems. Only one question was added to H-BRFSS about lifetime engagement; thus, we do not have information on the age at which respondents commenced (or stopped) spearfishing or the intensity of their engagement over time. These are important topics for future research. Moreover, the subjective nature of response categories for the outcome variable introduces interpretational variability, potentially impacting the precision and consistency of the data on this measure. Responses may also have been subject to recall bias, particularly among older respondents with longer lifetimes within which to judge the frequency of their engagement in spearfishing.

The study's cross-sectional design restricts the establishment of causal relationships. Instead, it provides a snapshot of the respondents' engagement in spearfishing at a specific point in time, preventing the examination of temporal sequences and causation. Because our study measured lifetime prevalence, our focus was not on causality, but on identifying if there are opportunities for interventions from a strengths-based perspective. While the study benefits from large sample size and careful consideration of complex weighting designs, the limitations described above should be considered when interpreting the findings in the broader context of public health research and discourse.

## Conclusion

While promoting physical activity is important, culturally relevant activities such as spear-fishing serve purposes beyond just fitness. They act as healing practices for both physical and emotional well-being [55]. Engaging with blue spaces and reviving culturally specific practices not only encourages physical activity, but is also directly associated with improved psycho-social well-being [11,30]. The findings of this study have valuable implications for guiding the implementation and design of interventions centered around culturally relevant physical activity. Native Hawaiian and Pacific Islander populations have been identified as at-risk and underserved [56], making them a key demographic for health promotion activities from a strengths-based and culturally rooted perspective. Similarly, other populations globally, including those who participate in spearfishing, require public health promotion congruent with their practice, values, and cultural strengths. Preventing chronic diseases is a key issue in health promotion for these groups, and integrating culture into health promotion strategies stands out as an effective means to address health inequities and poses a promising upstream approach to disease management [5]. Spearfishing, with the multifaceted aspects of health promotion in physical activity, nutrition, community-building, and engagement with nature, offers an important and understudied means to this goal.

## Supporting information

**S1 File. Variable Categorization.**
(DOCX)

**S2 File. Limited Dataset.**
(CSV)

**S1 Table. Prevalence of spearfishing (Often/Very Often), by demographic and health characteristics, among participants in the 2019 and 2020 H-BRFSS.**
(DOCX)

## Acknowledgments

We would like to thank David Sakoda at the Hawai'i Department of Aquatic Resources for his review of this paper and comments and feedback about the appropriateness of the conclusions as a spearfisherman himself and fisheries manager. We would like to thank Lance Ching from the Hawai'i Department of health for his professional support of this project, specifically providing input on the proper reporting of data from the Hawai'i Behavioral Risk Factor Surveillance System.

## Author contributions

**Conceptualization:** Yan Yan Wu, Tetine Lynn Sentell, Tonya Lowery St. John, Simone Schmid, Catherine McLean Pirkle.

**Data curation:** Tetine Lynn Sentell, Tonya Lowery St. John, Simone Schmid, Catherine McLean Pirkle.

**Formal analysis:** Lauryn Hansen, Yan Yan Wu.

**Funding acquisition:** Tetine Lynn Sentell, Catherine McLean Pirkle.

**Investigation:** Lauryn Hansen, Yan Yan Wu.

**Methodology:** Lauryn Hansen, Yan Yan Wu, Mika Thompson, Catherine McLean Pirkle.

**Project administration:** Yan Yan Wu, Tetine Lynn Sentell, Catherine McLean Pirkle.

**Resources:** Lauryn Hansen.

**Supervision:** Yan Yan Wu, Tetine Lynn Sentell, Catherine McLean Pirkle.

**Visualization:** Lauryn Hansen.

**Writing – original draft:** Lauryn Hansen.

**Writing – review & editing:** Lauryn Hansen, Tetine Lynn Sentell, Mika Thompson, Tonya Lowery St. John, Simone Schmid, Catherine McLean Pirkle.

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
