## [Decision Letter · Decision Letter 0]

19 Dec 2024

PONE-D-24-30068Spearfishing and public health promotion: A cross-sectional analysis of the Hawaiʻi Behavioral Risk Surveillance System SurveyPLOS ONE

Dear Dr. Pirkle,

Thank you for submitting your manuscript to PLOS ONE. After careful consideration, we feel that it has merit but does not fully meet PLOS ONE’s publication criteria as it currently stands. Therefore, we invite you to submit a revised version of the manuscript that addresses the points raised during the review process. The specific comments can be found at the bottom of this email.

We look forward to receiving your revised manuscript.

Kind regards,

Christopher A. Lepczyk

Academic Editor

PLOS ONE

**Journal Requirements:**

Funding: This work was supported by the Hawaiʻi Department of Health, Chronic Disease Prevention & Health Promotion Division, through a contract with the University of Hawai‘i at Mānoa. CMP and TLS were the recipients of the award. No commercial companies were involved in funding the work; no individual received funding from commercial companies for the purpose of this project. 

Funder role: The Hawaiʻi Department of Health is responsible for the Hawai'i Behavioral Risk Factor Surveillance System. CMP and TLS successfully negotiated with the Hawaiʻi Department of Health to include a question on spearfishing engagement during the 2019 and 2020 waves. Per national standards for the Behavioral Risk Factor Surveillance System, data collected from this system are available to the public following proper data request protocols. This includes the question on spearfishing engagement. The views expressed in the paper are the sole content of the authors and do not represent the official views of the Hawaiʻi Department of Health or the University of Hawai‘i. 

We would like to thank David Sakoda at the Hawaiʻi Department of Aquatic Resources for his review of this paper and comments and feedback about the appropriateness of the conclusions as a spearfisherman himself and fisheries manager. We would also like to thank Lance Ching at the Hawaiʻi Department of Health for his full support of this project, given that spearfishing had never been examined as a health topic in the state before. 

Funding: This work was supported by the Hawaiʻi Department of Health, Chronic Disease Prevention & Health Promotion Division, through a contract with the University of Hawai‘i at Mānoa. CMP and TLS were the recipients of the award. No commercial companies were involved in funding the work; no individual received funding from commercial companies for the purpose of this project. 

Funder role: The Hawaiʻi Department of Health is responsible for the Hawai'i Behavioral Risk Factor Surveillance System. CMP and TLS successfully negotiated with the Hawaiʻi Department of Health to include a question on spearfishing engagement during the 2019 and 2020 waves. Per national standards for the Behavioral Risk Factor Surveillance System, data collected from this system are available to the public following proper data request protocols. This includes the question on spearfishing engagement. The views expressed in the paper are the sole content of the authors and do not represent the official views of the Hawaiʻi Department of Health or the University of Hawai‘i. 

6. We note that there is identifying data in the Supporting Information file "Table S1". Due to the inclusion of these potentially identifying data, we have removed this file from your file inventory. Prior to sharing human research participant data, authors should consult with an ethics committee to ensure data are shared in accordance with participant consent and all applicable local laws.

-Location data

**Additional Editor Comments:**

Associate Editor:

Note: please include line numbers as it was very difficult to denote where to engage in edits without them (or page numbers).

Specific comments.

1. Where you use citation 13, I would note that to suggest any culture engaged in management that was resilient to fish and wildlife populations is unlikely. Since humans arrive in Hawaii there have been many species lost on land and it is likely that there was impacts in the ocean as well. I would suggest toning statements down a bit where we lack strong data to demonstrate best management practices. Fishing is and was culturally important and I think that is the main item to focus on.

2. For the hypotheses, I would suggest using active voice and stating ‘We hypothesized…” Also, to strengthen your hypothesis I would suggest providing literature citations or deductive reasoning. In the case of hypothesis 1, this felt a bit like a strawman I and think it needs more work. Also, both hypotheses would be stronger if you had a priori predictions that follow them.

3. Correlates. Are the variables you included modified in any way or are these how they are collated when you get them? For instance, is age binned or did you bin it? If you did modify any of the variables, please note it and why you did it (this situation also might determine if any different analyses are needed).

4. Missing a comma in ‘and greater) island of’ which should be ‘and greater), island of.’

5. Statistical Analysis. Please unify paragraphs into fewer as a paragraph should have a minimum of three sentences.

6. When the word ‘this’ is used to begin a sentence it needs an object following it that connects back to previous sentence. Please correct throughout the ms.

7. Statistical models. I did not understand why just three different models were used. What was the a priori reason for doing these and not other ones or not looking at a best model or model set? In addition, what are you using to define statistical significance in your analysis? Finally, I would suggest noting what a prevalence ratio is and how to interpret it as many readers are not going to understand what is presented in the tables.

8. Results. I would suggest avoiding starting out a sentence like ‘Table 1…’ Rather, please write out the main results and put the table or figure that describes them in more detail in parentheses. You want to state the items here that the reader should learn and not send them right to the tables.

9. Discussion. I would suggest beginning your Discussion by indicating whether or not you found support for your hypotheses.

Reviewers' comments:

Reviewer's Responses to Questions

**Comments to the Author**

1. Is the manuscript technically sound, and do the data support the conclusions?

Reviewer #1: Yes

Reviewer #2: Yes

2. Has the statistical analysis been performed appropriately and rigorously? 

Reviewer #1: I Don't Know

Reviewer #2: Yes

3. Have the authors made all data underlying the findings in their manuscript fully available?

Reviewer #1: Yes

Reviewer #2: Yes

4. Is the manuscript presented in an intelligible fashion and written in standard English?

Reviewer #1: Yes

Reviewer #2: Yes

5. Review Comments to the Author

**Reviewer #1: ** A well summarized paper with appropriate references. The goal of increasing the recognition of spearfishing in light of overfishing is concerning but thoughtfully addressed in the paper. There was just a slight interesting twist in the logic however knowing that this paper is trying to bring attention to the sport of spearfishing to address public health yet most of the respondents that actively spearfish are currently overweight. If I misunderstood that, perhaps dissecting that a bit further would be helpful.

**Reviewer #2:**  Review

PLOS

Spearfishing

Spearfishing is a culturally important activity, and could proffer health benefits. In Hawaiʻi, there is no data on the prevalence of this fishing activity in state datasets. This study analyzes two years of data acquired through a question related to spear fishing activity added to the Hawaii Behavioral Risk surveillance System Survey. The study finds interesting details related to fishing behavior. The results are relevant to health promotion activities (and, I imagine, aquatic conservation). The study is well motivated, and the paper is clearly written. The methods are explained in sufficient detail. The results are presented in tables that are easy to interpret. The discussion, study limitations, and conclusion follow from the analysis.

I recommend publishing.

I do have some minor comments for the authors to consider:

“It is believed that during the peak of the Hawaiian Kingdom, nearly every Hawaiian resident participated in some form of fishing.” The way this is phrased begs for a citation.

Not sure about only putting the citation in the 2nd sentence: “One study on the island

of Hawaiʻi documented that 92% of fish caught remained within a 75-km radius of where they

were caught, with half being kept and another third being given away. Additionally, a full fifth of

all fish caught was used for cultural events such as birthdays, weddings, and funerals (16).”

You group your population two ways (ʻsometimes,’ ‘often’ or ‘very often’ during their lifetimes with those who reported participating ‘never’ or ‘almost never’ and ‘often’ or ‘very often’). Of course, it is completely subjective what these terms mean, but do you think that someone who answers “sometimes” is really getting the health benefits? Can you justify why you chose the broader grouping, i.e., including sometimes (other than then you have a larger number)?

For those you are not statisticians, you might explain a tiny bit more what the statistics mean (for instance, “the PR indicates the change in prevalence in spearfishing associated with the correlate (e.g., sex, age, health status)”. A little more explanation of crude vs. adjusted, and the Poisson regression would make the paper a great teaching tool.

6. PLOS authors have the option to publish the peer review history of their article (what does this mean? ). If published, this will include your full peer review and any attached files.

**Do you want your identity to be public for this peer review?** For information about this choice, including consent withdrawal, please see our Privacy Policy .

Reviewer #1: No

Reviewer #2: **Yes: ** Kirsten Oleson

---

## [Author Response · Author response to Decision Letter 1]

23 Jan 2025

Dear Editors and Reviewers,

We greatly appreciate the opportunity for revision, and for the time spent by each reviewer in helping improve our submission. Below, you will find a point-by-point response to each reviewer comment that we received.

Journal Requirements

1. Manuscript Style: The following adjustments per the PLOS one style requirements addressed in our overarching response have been made.

a. Separation of manuscript body and title page

Response: The manuscript title page was placed in a separate document from the manuscript body and uploaded separately.

b. Page and line numbers

Response: Beginning with the abstract, page numbers were added to the bottom left corner, and line numbers were added along the left-hand side of the manuscript body. Line numbers were added to the title pages & abstract separately to mirror the examples in the PLOS ONE style templates.

2. Funder Role

Response: As requested, the statement “There was no additional external funding received for this study” was added to our statement after “no individual received funding from commercial companies for the purpose of this project”

There is no grant number associated with the funding. The funding that supported this work was a contract, not a grant. This is more clearly articulated in the Funding statement.

3. Acknowledgments Section

Response: Thank you for the recognition of our acknowledgments statement. We see the note requesting no detailing of any funding in the acknowledgements. To clarify that those acknowledged are not funders, but rather professional support, we reformatted the sentence as follows:

“We would like to thank Lance Ching from the Hawaiʻi Department of health for his professional support of this project, specifically providing input on the proper reporting of data from the Hawaiʻi Behavioral Risk Factor Surveillance System.”

4. Data Availability statement

The data have been made available in the supporting information.

5. Ethics Statement

Response: Thank you for pointing out that we had incorrectly placed our ethics statement at the end of our manuscript. It is now in the final subheading in our methods section at Line 229

6. Identifiable Participatory Information in Supplemental Table 1

Response:

The data we used are publicly available. The Behavioral Risk Factor Surveillance (BRFSS) System is an annual, nationally representative telephone survey of United States adults. A limited dataset is made publicly available for secondary data analysis with minimum cell size restrictions to ensure participant privacy and estimate validity. https://www.cdc.gov/brfss/data_documentation/index.htm

The BRFSS does not collect participant names, street addresses or date of birth or other identifying information.

https://health.hawaii.gov/brfss/participation-in-hawaii-brfss/

In conducting our analyses using BRFSS data, we followed all applicable rules regarding the reporting of results and did so in consultation with and review by the Hawaiʻi Department of Health, which houses and manages Hawaiʻi BRFSS data. The data provided in the file Spearfishing Data.csv are from the national BRFSS, except for the Spearfishing variable (outcome and described in the paper) and race/ethnicity variable (these are coded specifically for Hawaiʻi given the unique make-up of the state). The Hawaiʻi specific data are also publicly available through the Hawaiʻi Department of Health. The results in S1_Table are measures of association (prevalence ratios), not frequencies. There are no cells in the table that could be combined so as to inadvertently identify a small group of participants.

7. Captions in supplemental tables

Response: Thank you for pointing out the proper caption for our supplemental information. The caption has been changed from Supplemental Table 1, to Table S1, with the associated descriptive text. We have also added the supplemental information section to the end of the manuscript on page 33:

Supporting Information

S1 Table. Prevalence of spearfishing (Often/Very Often), by demographic and health characteristics, among participants in the 2019 and 2020 H-BRFSS

S1 File: Variable Categorization

S2 File: Limited Dataset

8. References List

Response: The citations listed below were added to support the definition of a prevalence ratio on line 205 on page 9 (#40) and the statement about Native Hawaiian demographics (#41) and the population size of Molokaʻi and Lānaʻi (#43), page 22, lines 342 and 348, respectively.

40. Gnardellis C, Notara V, Papadakaki M, Gialamas V, Chliaoutakis J. Overestimation of Relative Risk and Prevalence Ratio: Misuse of Logistic Modeling. Diagnostics. 2022 Nov 17;12(11):2851.

41. U.S. Census Bureau. data.census.gov. [cited 2025 Jan 2]. Lanai CCD, Maui County, Hawaii - Census Bureau Profile. Available from: https://data.census.gov/profile/Lanai_CCD,_Maui_County,_Hawaii?g=060XX00US1500992070

42. Office of Hawaiian Affairs. Island Community Report: Molokaʻi. Honolulu, HI: Office of Hawaiian Affairs; 2024 [cited 2025 Jan 2]. Available from: https://www.oha.org/wp-content/uploads/2024-Molokai-Island-Community-Report.pdf

To the best of our knowledge, the reference list reflects the complete scope of citations from the paper.

Response(s) to Editor

Associate Editor

C: Where you use citation 13, I would note that to suggest any culture engaged in management that was resilient to fish and wildlife populations is unlikely.

R: The sentence corresponding to citation 13, “At the same time, traditional management systems supported healthy and resilient marine ecosystems (13)” has been removed.

C: For the hypotheses, I would suggest using active voice and stating ‘We hypothesized…” Also, to strengthen your hypothesis I would suggest providing literature citations or deductive reasoning

R: The hypothesis has been changed from “it is hypothesized” to “we hypothesize” to reflect this request. Additionally, the historical accounts of spearfishing in Hawaiʻi and related studies on hula and paddling prevalence in Hawaiʻi have now been added and cited to substantiate the hypothesis. This now reads: “Building on findings of higher lifetime engagement in culturally relevant activities such as hula and paddling among Native Hawaiians and Other Pacific Islanders (10), and considering the extensive history of fishing in the Hawaiian Islands, we hypothesize a high prevalence of spearfishing engagement (13), particularly among Native Hawaiian and Pacific Islander communities. Furthermore, we hypothesize that participation would vary based on sociodemographic and health factors, with younger individuals and those in better health being more likely to participate.” See page 6, lines 138 to 140.

C: Correlates. Are the variables you included modified in any way or are these how they are collated when you get them? For instance, is age binned or did you bin it? If you did modify any of the variables, please note it and explain why you did it (this situation also might determine if any different analyses are needed).

R: A supplemental file has been made to detail how the various correlates were coded and provides explanations for coding decisions. The reader is directed to the supplemental file at line 187.

C: Missing a comma in ‘and greater) island of’ which should be ‘and greater), island of.’

R: Thank you for catching this. The comma has been added.

C: Statistical Analysis. Please unify paragraphs into fewer as a paragraph should have a minimum of three sentences.

R: The first two paragraphs in the statistical analyses section have been combined on pages 9-10.

C: When the word ‘this’ is used to begin a sentence it needs an object following it that connects back to previous sentence. Please correct throughout the ms.

RL The authors believe the sentence being referenced was: "This observation is reinforced by the associations between obesity and difficulty walking and climbing stairs with lifetime spearfishing engagement.” The paragraph (lines 368-373) has been revised for clarity and now reads: “The prevalence of lifetime spearfishing engagement showed minimal variation across age groups, with only a five percent difference between the highest reported engagement (26.9% among individuals aged 25-34 years) and the lowest (21.6% among those > 65 years). The findings suggest that spearfishing may serve as an accessible activity for people of various ages at some point in their lifetime. The idea of spearfishing as an accessible activity is further supported by its associations with difficulty walking or climbing stairs.”

C: Statistical models. I did not understand why just three different models were used. What was the a priori reason for doing these and not other ones or not looking at a best model or model set? In addition, what are you using to define statistical significance in your analysis? Finally, I would suggest noting what a prevalence ratio is and how to interpret it as many readers are not going to understand what is presented in the tables.

R: Thank you for your feedback regarding the clarity of our analysis. The three models were selected to incrementally adjust for potentially confounding factors, allowing the reader to consider how these factors each influence the relationship with spearfishing engagement and health outcomes.

• Model 1 adjusted only for demographic characteristics (age, sex, race/ethnicity) to identify baseline associations and disparities in health outcomes across demographic groups.

• Model 2 adjusted for health behaviors (meeting physical activity guidelines, smoking status, etc.), accounting for individual-level behaviors that might confound the relationship between health outcomes and spearfishing engagement.

• Model 3 further adjusts for socioeconomic factors (e.g., income and education), which are critical determinants of health and access to activities like spearfishing.

This stepwise approach reflects standard practice in cross-sectional studies, including the paper published by Sentell et al (2023). Our modeling strategy is consistent with public health surveillance, which seeks to describe frequencies of events or characteristics rather than to ascribe causality. In other words, survey research such as this, which examines prevalence and correlates associated with prevalence in a population, applies different modeling strategies from more etiological research examining hypothesized risk factors and which often uses a modeling strategy that seeks the most parsimonious model to describe an exposure-outcome relationship.

Regarding statistical significance, we are considering a result statistically significant if the p-value is 0.05 or less. This has been noted on Page 10, line 222: “Results were deemed statistically significant if the p-value was 0.05 or less.”

A prevalence ratio (PR) is also a commonly used measure in epidemiology that assesses the strength of the association between an exposure/characteristic and an outcome. It compares the prevalence (cases/total population) of an outcome in those who are exposed to those who are not unexposed. For example, a PR of 1.5 indicates that the outcome is 1.5 times as likely or 50% more likely in the exposed group compared to the reference group. Conversely, a PR of 0.75 would indicate that the outcome is 25% less likely in the exposed group. To improve reader understanding, the definition of a prevalence ratio—and respective citation—has been added to line 203 in the Statistical Analysis section of the methods: “ A prevalence ratio (PR) is commonly used to assess the strength of the association between an independent variable and outcome of interest (40). It compares the prevalence of an outcome in those who have a certain characteristic to those who do not.”

Sentell T, Wu YY, Look M, Gellert K, Lowery St John T, Ching L, et al. Culturally Relevant Physical Activity in the Behavioral Risk Factor Surveillance System in Hawai’i. Prev Chronic Dis. 2023 May 25;20:E43.

Gnardellis C, Notara V, Papadakaki M, Gialamas V, Chliaoutakis J. Overestimation of Relative Risk and Prevalence Ratio: Misuse of Logistic Modeling. Diagnostics. 2022 Nov 17;12(11):2851.

C: Results. I would suggest avoiding starting out a sentence like ‘Table 1…’ Rather, please write out the main results and put the table or figure that describes them in more detail in parentheses. You want to state the items here that the reader should learn and not send them right to the tables.

R: The paragraph was rephrased to avoid starting with “Table.” See page 11.

“An estimated 25.5% of the adult population of Hawaiʻi engaged in spearfishing sometimes, often, or very often during their lifetimes. The weighted population size of those engaging with spearfishing during their lifetimes was 226,814 people, with 89,750 estimated to have done so often or very often (Table 1). Detailed population characteristics of the sample are provided in Table 1.

The crude and adjusted PRs for the association of each correlate with lifetime spearfishing engagement were calculated across all models (Table 2).” See page 13.

Of note, due to the extensive nature of the data, detailed population characteristics encompassing the 21 demographic correlates are summarized in Table 1. Including all results in the text would disrupt the flow of the narrative, but the tables provide comprehensive information for readers seeking a deeper understanding.

C: Discussion. I would suggest beginning your Discussion by indicating whether or not you found support for your hypotheses.

R: The first paragraph of the discussion, as well as the first sentence of the second paragraph of the discussion, were rephrased to explicitly state whether or not we found support for our hypothesis. The respective sentences read: “The results of this study support this hypothesis, with 1 in 4 Hawaiʻi residents reporting engaging in spearfishing during their lifetime.” And, “In accordance with our hypothesis, we also found that demographic and health-related correlates were associated with lifetime spearfishing engagement”

Reviewer 1:

C. The manuscript is technically sound, and the data supports the conclusions.

R. We appreciate your comment, thank you.

C: The authors have made all data underlying the findings in their manuscript fully available.

R: Thank you.

C: The manuscript is presented in an intelligible fashion and written in standard english

R: We appreciate this acknowledgement, thank you.

C: A well summarized paper with appropriate references. The goal of increasing the recognition of spearfishing in light of overfishing is concerning but thoughtfully addressed in the paper.

R: Thank you for the positive comments and for acknowledging our efforts to address the topic of overfishing thoughtfully and with sensitivity.

C: There was just a slight interesting twist in the logic however knowing that this paper is trying to bring attention to the sport of spearfishing to address public health yet most of the respondents that actively spearfish are currently overweight. If I misunderstood that, perhaps dissecting that a bit further would be helpful.

R: We acknowledge that the relationship between spearfishing and weight status warrants further consideration. As detailed in our analysis, the unadjusted models show a higher prevalence of overweight or obesity among individuals who reported lifetime engagement in spearfishing. However, this association largely disappeared after adjusting for demographic variables, including race/ethnicity and island of residence. This is explained on page 20, lines 294-301 and pasted below for ease of review.

“There was a higher prevalence of spearfishing engagement among those who were overweight or obese in the unadjusted model. However, the association became non-significant for overweight (PR: 1.05, 95%CI: 0.95-1.15) and marginally significant for obese individuals (PR:1.10, 95%CI: 0.99-1.22) in Model 3, which statistically adjusted for demographic characteristics, health behaviors, and

---

## [Editor Report · Decision Letter 1]

29 Jan 2025

Spearfishing and public health promotion: A cross-sectional analysis of the Hawaiʻi Behavioral Risk Surveillance System Survey

PONE-D-24-30068R1

Dear Dr. Pirkle,

We’re pleased to inform you that your manuscript has been judged scientifically suitable for publication and will be formally accepted for publication once it meets all outstanding technical requirements.

Kind regards,

Christopher A. Lepczyk

Academic Editor

PLOS ONE
---

## [Editor Report · Acceptance letter]

PONE-D-24-30068R1

PLOS ONE

Dear Dr. Pirkle,

I'm pleased to inform you that your manuscript has been deemed suitable for publication in PLOS ONE. Congratulations! Your manuscript is now being handed over to our production team.

Kind regards,

on behalf of

Dr. Christopher A. Lepczyk

Academic Editor

PLOS ONE